# Fully Automated Macro- and Microfluidic Production of [^68^Ga]Ga-Citrate on mAIO^®^ and iMiDEV^TM^ Modules

**DOI:** 10.3390/molecules27030994

**Published:** 2022-02-01

**Authors:** Olga Ovdiichuk, Emilie Roeder, Sébastien Billotte, Nicolas Veran, Charlotte Collet

**Affiliations:** 1Nancyclotep, Molecular Imaging Platform, F-54500 Vandoeuvre-les-Nancy, France; e.roeder@nancyclotep.com (E.R.); seb.billotte@laposte.net (S.B.); 2CHRU-Nancy, Service de Medecine Nucléaire, Pôle Pharmacie, F-54000 Nancy, France; n.veran@chru-nancy.fr; 3Université de Lorraine, Inserm, IADI, F-54000 Nancy, France

**Keywords:** [^68^Ga]Ga-citrate, automated radiosynthesis, quality control, radiopharmaceuticals

## Abstract

^68^Ga-radionuclide has gained importance due to its availability via ^68^Ge/^68^Ga generator or cyclotron production, therefore increasing the number of ^68^Ga-based PET radiopharmaceuticals available in clinical practice. [^68^Ga]Ga-citrate PET has been shown to be prominent for detection of inflammation/infection of the musculoskeletal, gastrointestinal, respiratory, and cardiovascular systems. Automation and comparison between conventional and microfluidic production of [^68^Ga]Ga-citrate was performed using miniAllInOne^®^ (Trasis) and iMiDEV™ (PMB-Alcen) synthetic modules. Fully automated procedures were elaborated for cGMP production of tracer. In order to facilitate the tracer approval as a radiopharmaceutical for clinical use, a new method for radiochemical identity determination by HPLC analysis to complement standard TLC radiochemical purity measurement was developed. The results showed higher radiochemical yields when using MCX cartridge on the conventional module mAIO^®^, while a PS-H+ cation exchanger was shown to be preferred for integration into the microfluidic cassette of iMiDEV™ module. In this study, the fully automated radiosynthesis of [^68^Ga]Ga-citrate using different synthesizers demonstrated reliable and reproducible radiochemical yields. In order to demonstrate the applicability of [^68^Ga]Ga-citrate, in vitro and in vivo studies were performed showing similar characteristics of the tracer obtained using macro- and microfluidic ways of production.

## 1. Introduction

Nowadays, the interest in radiopharmaceuticals based on gallium-68 is increasing due to easy access to the gallium radioisotope. Indeed, ^68^Ga is obtained by elution of ^68^Ge/^68^Ga generators with hydrochloric acid or via a cyclotron-based production to give ^68^GaCl_3_. The availability of PET devices and ^68^Ga has grown the interest in imaging with [^68^Ga]Ga-citrate since the past decade.

Gallium-68 is a radioisotope with a short half-life (68.8 min), and it is a positron emitter allowing its use for positron emission tomography (PET) imaging. ^68^Ga-radiochemistry is coordination chemistry consisting in labelling ^68^Ga^3+^ onto chelating agents such as, DOTA, NOTA, or HBED coupled to targeting agents to image specific receptors (e.g., [^68^Ga]Ga-DOTATOC for neuroendocrine tumour, [^68^Ga]Ga-PSMA-11 for prostate cancer, etc.).

The first radiopharmaceutical based on citrate used for diagnosis in clinical applications was [^67^Ga]Ga-citrate. It has been used for imaging inflammation and infection by SPECT for more than 40 years, in particular for identifying bone and joint infections [1,2]. However, this radiopharmaceutical is less and less used due to high radiation dose and the long half-life of gallium-67 (78.3 h). Most studies are focused on Ga as an iron biomimetic to diagnose prosthetic joint infections by PET imaging [3,4] (two clinical trials terminated, another one not yet recruiting in clinicaltrials.gov). Indeed, ^67^Ga, ^68^Ga, and iron have similar uptake mechanisms via the binding to transferrin or siderophores followed by the delivery of complexes to cells or specific receptors [5].

Furthermore, the diagnostic value of [^68^Ga]Ga-citrate over [^18^F]FDG PET is not fully documented. [^18^F]FDG imaging is a method of interest for the diagnosis of infection or inflammation, with a very good availability of the last. Nevertheless, [^18^F]FDG may have some limitations, as it might be falsely positive in some cases [6].

Other on-going clinical applications are focused on the utility of [^68^Ga]Ga-citrate imaging on tuberculosis [7,8], fever of unknow origin [4], and bone infections [9,10,11,12].

More recently applications outside the field of infectious diseases such as sarcoidosis [13] and gliomas [14] started being investigated.

In the literature, the radiosynthesis of [^68^Ga]Ga-citrate is often described using a manual method consisting in elution of ^68^GaCl_3_ from a ^68^Ge/^68^Ga generator which is then added to a solution of sodium citrate, under stirring, with or without heating [3,5,6,15,16]. However, to increase the reproducibility of the radiosynthesis and to decrease the radiation exposure of the operator, the use of an automated synthesis module is preferred. First, the radiosynthesis of [^68^Ga]Ga-citrate was described by Rizello et al. using the commercial semi-automatic labelling module F-CON from Eckert and Ziegler [17]. The radiosynthesis strategy consists in the prepurification of a Ga-68 eluate from potential metal traces by the Zhernosekov method (HCl/Acetone) [18] on a Accell Plus CM cationic cartridge, followed by a labelling in a vessel for 5 min at 70 °C. Recently, Ugur et al. used this strategy to produce [^68^Ga]Ga-citrate on a Scintomics GmbH GRP module 4 V synthesis module by a cationic method of ^68^Ga-prepurification (PS-H+ cartridge), followed by labelling in a reactor during 10 min at 70 °C [19]. Overall, the described methods suffer from a potential metal ion impurity breakthrough, the removal of the organic solvent, and the related additional quality control analysis or the prolonged synthesis times because of the heating step.

In 2013, Jensen et al. was the first to propose [^68^Ga]Ga-citrate radiosynthesis on a strong cation exchange (SCX) cartridge with a commercially available Modular-Lab^®^ from a PharmTrace module using a disposable cassette [20]. This strategy, consisting in a Ga-68 trapping–desorption with [^68^Ga]Ga-citrate formation, allows all of the drawbacks of the previous methods to be avoided.

As standard method for radiochemical purity control of this radiopharmaceutical, radio-TLC is commonly used, and only few researchers have described the determination of radiochemical purity using HPLC [5,19,21].

Herein, we describe the production of [^68^Ga]Ga-citrate on two commercial synthesizers. The first one is the macrofluidic miniAIO^®^ (Trasis), and it has been used for production of numerous ^68^Ga-radiopharmaceuticals such as [^68^Ga]Ga-PSMA-11 [22,23] and [^68^Ga]Ga-NODAGA-RGD [24]. The second one is iMiDEV™ (PMB Alcen), a new commercially available synthesizer using microfluidic technology for radiopharmaceutical production [25]. The same simplified radiosynthesis strategy was successfully implemented onto both radiosynthesis modules.

For routine quality control of this radiopharmaceutical, a high-performance liquid chromatography (HPLC) identification method was developed, in view of the stringent rules for on-site GMP synthesis of radiopharmaceuticals adopted in most European countries. Additional aspects were considered in order to validate the [^68^Ga]Ga-citrate production, including the metal ion contaminants, in vitro stability, and binding and in vivo biodistribution of the radiopharmaceutical.

## 2. Results

For the purpose of producing [^68^Ga]Ga-citrate with high radiochemical yield and purity, an optimization of the radiosynthesis was performed. The radiosynthesis strategy was adapted from the method described by Jensen et al. (Figure 1) [20]. Our method included the trapping of ^68^Ga^3+^ cations on a solid support with cation exchange beads, the wash of beads, and the elution of the Ga-68 with sodium citrate solution, followed by a final sterile filtration of the [^68^Ga]Ga-citrate in the product vial.

### 2.1. Radiosynthesis of [^68^Ga]Ga-Citrate

#### 2.1.1. Optimization of [^68^Ga]Ga-Citrate Radiosynthesis on mAIO^®^

The radiosynthesis of [^68^Ga]Ga-citrate on the mAIO^®^ module included all steps described above, starting from the eluate trapping and ending with [^68^Ga]Ga-citrate formulation.

For optimization two parameters were evaluated: the nature of cation exchange solid support and the concentration of sodium citrate solution.

Determination of adapted cation exchange solid support

Different types of commercially available solid supports were tested to evaluate their capacity to trap and to desorb ^68^Ga^3+^ with a sodium citrate solution. Results of this investigation are summarized in Table 1.

Not surprisingly, cartridges with the worst trap–release efficiency were the weak cation exchangers WCX and Accell plus CM containing carboxylate function, while MCX and PS-H+, grafted with aromatic sulfonic acid functions, showed the best trap–release efficiency with 136 mM sodium citrate solution as an eluent (up to 96%). Despite the high trapping capacity of strong cation exchangers SCX and AG 50W-X8, these beads showed a low ^68^Ga^3+^ elution. The activity was almost completely eluted from the MCX and PS-H+ cartridges, which made them well-adapted for further experiments.

Determination of optimal concentration of sodium citrate

The two best cartridges (MCX and PS-H+) were chosen to evaluate the impact of the concentration of the sodium citrate solution on the [^68^Ga]Ga-citrate formation and quality. The concentrations of sodium citrate were chosen to be compatible with clinical use.

Thus, two sodium citrate concentrations were chosen based on concentrations presented in commercially available solutions used for intravenous injections as anticoagulant. The first one, ACD-A, is a mixture of sodium citrate/citric acid/glucose (74.8 mM/38 mM/123.6 mM respectively), and the second one is a trisodium citrate (TSC) 4% solution containing 136 mM of sodium citrate. We decided to evaluate these two concentrations (74.8 and 136 mM) and one of the pattern solutions (ACD-A). Powder of sodium citrate of Ph. Eur. quality and NaCl 0.9% were used as an eluent preparation allowing an isotonic solution which could be directly used for injection to be obtained.

Using two different cation exchange-based cartridges, an influence of the eluting solution on the [^68^Ga]Ga-citrate RCY was observed. Both solutions of sodium citrate (74.8 mM and 136 mM in NaCl 0.9%) provided better results compared with ACD-A solution containing a mixture of citric acid and sodium citrate solution (Table 2).

For PS-H+, the elution with the 136 mM sodium citrate solution provided higher radiochemical yield compared with the 74.8 mM sodium citrate solution. In the case of the MCX cartridge, both 74.8 and 136 mM concentrations allowed [^68^Ga]Ga-citrate with comparably high radiochemical yields and with a radiochemical purity >99% to be obtained.

Radiosynthesis automation

The optimization-based production of [^68^Ga]Ga-citrate radiopharmaceutical was performed on the mAIO^®^ module using the MCX cartridge and the 136 mM sodium citrate solution in NaCl 0.9%. The graphical representation of synthetic layout is depicted in Figure 2. The in-house assembled manifold installed into the mAIO^®^ module is completely customizable with a commercial cation exchange cartridge and different reagents in positions that could be modified if needed. The above optimizations were combined and implemented into the corresponding automated sequence. This sequence included all main steps, processes, and operations needed to perform an automated [^68^Ga]Ga-citrate production of the mAIO^®^. The MCX cartridge was placed in position 10, bag C with NaCl 0.9% for washing in position 6, and a vial containing the 136 mM sodium citrate solution in NaCl 0.9% A in position 4. The radiosynthesis time was 11 min from the beginning of elution to the end of the delivery of the formulated product, with a radiochemical yield of 95.8 ± 0.8 % (*n*= 32) decay corrected to the SOE (start of elution). The radiation detector placed on the mAIO^®^ modules allowed the monitoring of the radiosynthesis (Appendix A).

#### 2.1.2. Optimization of [^68^Ga]Ga-Citrate Radiosynthesis on iMiDEV™

Based on the optimization results found for the conventional production of [^68^Ga]Ga-citrate, the synthetic strategy described in Figure 1 was translated on the microfluidic iMiDEV™ cassette (Figure 3). Microchambers R1 and R3 with a same volume of 50 µL were chosen as reaction chambers. These chambers are recommended for filling with SPE beads and could be both used for the radionuclide trapping step. This should allow two productions of [^68^Ga]Ga-citrate radiopharmaceutical using the same cassette to be performed.

Two types of beads, MCX and PS-H+, providing the best trapping–elution ratio for [^68^Ga]Ga-citrate in conventional radiosynthesis, were integrated into the R1 and R3 chambers of the [^68^Ga]Ga-citrate cassette. The trapping efficiency on both chambers and using either MCX or PS-H+ type of beads had been shown to exceed 95% (Table 3). The trapping time depended on the volume of the initial activity (20–30 s for 1 mL and up to 2 min for 5 mL of ^68^GaCl_3_). No significant influence of the chamber type (R1 or R3), the type of beads (MCX or PS-H+), the starting activity (70–420 MBq), or the volume of initial ^68^GaCl_3_ (1–5 mL) was detected on the trapping efficiency.

The trapping step was followed by a wash of the cationic cartridge with 1 mL of saline solution. ^68^Ga^3+^ desorption with formation of [^68^Ga]Ga-citrate was performed using 4 mL 136 mM sodium citrate solution as an eluent based on the optimization results described above for the macrofluidic production. The recovery of activity was up to 91% for all cases.

No significant difference was observed in trap–release of ^68^Ga^3+^ between MCX and PS-H+ beads; therefore, it was decided to use and to compare both types of cassettes for the [^68^Ga]Ga-citrate production.

Thus, [^68^Ga]Ga-citrate on the R1 was obtained in 90.1 ± 3.1% yield when filled with PS-H+ cation exchanger and with an 84.9 ± 4.8% yield when MCX was used. A slight decrease of about 3–4% of the radiochemical yield occurred when using the R3 chamber filled either with PS-H+ or MCX beads (Table 4). The radiochemical purity of [^68^Ga]Ga-citrate in radiolabelling performed on R1 filled with PS-H+ beads was 98.7 ± 1.2%, and in the case of production on cassettes with MCX in R1, it was 98.8 ± 3.2%. Surprisingly, a significant degradation of the radiochemical purity was observed for the final product obtained on R3 filled with MCX resin in contrast with [^68^Ga]Ga-citrate of a high purity obtained on the same chamber filled with PS-H+ beads.

Regarding higher radiochemical yields and radiochemical purity while using a PS-H+ cation exchanger, it was decided to perform all quality control validation tests on cassettes filled with this type of beads in the R1 chamber.

Automation of radiosynthesis

The fully automated production of [^68^Ga]Ga-citrate was accomplished by creating the corresponding sequence consisting in necessary actions to perform the steps described above (Appendix A). The vial A containing NaCl 0.9% for washing and the vial B with a 136 mM sodium citrate solution in NaCl 0.9% were loaded on the cassette prefilled with PS-H+ in the R1 chamber. The radiosynthesis time was 10 min from the beginning of trapping to the end of the delivery of the formulated product in high radiochemical yield and purity.

The radiation detectors placed on the R1 chamber recorded the activity signal and allowed the [^68^Ga]Ga-citrate radiosynthesis (Appendix A) to be monitored.

Residual activity distribution

In order to visualize the distribution of the residual activity in the cassette, a Cerenkov luminescence imaging (CLI) was applied after the completion of the production. In all experiments, no activity was detected in the microfluidic channels of the bottom part of the cassette; therefore, only images of the top part are presented hereafter.

CLI after synthesis on R1 showed that the residual activity stayed trapped on the solid support (Figure 4).

### 2.2. Quality Control of Metal Ions Impurities

In order to discard an excess of Zn(II) that forms with the decay of ^68^Ga and accumulates in the ^68^Ge/^68^Ga generator system, a preventive elution at least 24 h prior the sample production was performed. Using the atomic absorption method (AAS), the concentrations of iron (from the sealing parts and acid impurities) and zinc (as a decay product) were determined in the ^68^GaCl_3_ obtained after the generator elution with 10 mL 0.1 M HCl and in the formulated [^68^Ga]Ga-citrate solutions obtained on both synthetic modules. (Table 5).

These results demonstrate that our radiosynthesis strategy allowed a considerable decrease in the Zn level in the final formulated solution in comparison with the Zn(II) concentration in the generator eluate. The radiosynthesis on the mAIO^®^ module using MCX solid support was revealed to be less efficient for zinc removal compared with the radiosynthesis on the iMiDEV^TM^ cassette filled with PSH+ beads. Very low concentrations of Fe were measured in all samples.

### 2.3. Development of Quality Control Procedure

#### 2.3.1. Radio-TLC

The literature review revealed a variety of mobile and stationary phases used for the measurement of radiochemical purity by the iTLC-SG method. In order to determine the analytical conditions providing the best resolution between ^68^Ga^3+^ and [^68^Ga]Ga-citrate, a series of tests was performed. The Rf values for each system are described in Table 6 and the radiochromatograms are presented in Appendix A.

The difference in retention factors while using the system described by Xu et al. was too low for quantification of peaks, potentially yielding an overlapping of ^68^Ga^3+^ and [^68^Ga]Ga-citrate. With the system described by Mirzaie and co-workers, the radiochromatogram obtained for ^68^Ga^3+^ did not allow both species to be quantified because ^68^Ga^3+^ presented two peaks—the first one of the free Ga-68 with a Rf = 0.2 and the second one with a Rf = 0.95 attributed to acetylated species of Ga-68.

The conditions described by Jensen et al. using a mixture of AcONa/AcOH in water as a mobile phase were considered to provide the best resolution between ^68^Ga^3+^ and [^68^Ga]Ga-citrate without any peak tailing. To evaluate a potential formation of colloidal [^68^Ga]Ga, an additional radio-TLC was performed with ACD formula A. The analyses using these two systems confirmed the quantitative formation of [^68^Ga]Ga-citrate radiopharmaceutical, and non-colloidal [^68^Ga]Ga was detected.

#### 2.3.2. Radio-HPLC Identification Methods Development

A new method for the identification of [^68^Ga]Ga-citrate by HPLC analysis was developed based on the HPLC quality control of trisodium citrate described by Levesque et al. [26]. Typical radio and UV chromatograms for batches of [^68^Ga]Ga-citrate produced using mAIO^®^ and iMiDEV™ synthesizers are shown in Figure 5. Free Ga-68 was injected under the same conditions, but no detection of ^68^GaCl_3_ was observed in 10 min analysis.

### 2.4. Validation of [^68^Ga]Ga-Citrate Production

The best conditions obtained for mAIO^®^ (MCX cartridge and sodium 136 mM citrate solution 4 mL) and for iMiDEV™ synthesizers (PS-H+ beads in R1 chambers and 136 mM sodium citrate solution) were used for the validation of [^68^Ga]Ga-citrate production.

Completed quality control experiments were realized on three consecutive batches realized on each radiosynthesis module, and all values are given as an average (Table 7).

### 2.5. Evaluation of the Stability In Vitro

The stability of [^68^Ga]Ga-citrate in formulated solution at room temperature was evaluated for up to 3 h by measuring the radiochemical purity of the samples using iTLC-SG analysis. The results demonstrated an appropriate stability (RCP > 98%) along this time period. The stability of [^68^Ga]Ga-citrate in serum for up to 1 h at 37 °C was also evaluated using iTLC-SG tests. In this case, the radiochemical purity value obtained at 1 h post-incubation was 82% (Appendix A).

### 2.6. In Vitro Characterization

The serum protein (SPB) and the blood cell bindings (BCB) of [^68^Ga]Ga-citrate were found to be 49.9 ± 1.6% and 66.8 ± 2.8%, respectively, demonstrating the high affinity of the radiopharmaceutical to complex transferrin and other siderophore proteins.

### 2.7. Physiological Biodistribution

The biodistribution of [^68^Ga]Ga-citrate solution was evaluated in vivo on healthy rats using 2 h dynamic acquisitions on a microPET (Inveon, Siemens, USA). ROI (regions of interest) were placed on various organs to obtain mean standard uptake values (SUV mean) by body weight (bw), presented in Table 8 (*n* = 3). In order to evaluate a potential difference between the two modules, a *t*-test was applied. No difference of any organ uptake was found between the syntheses of [^68^Ga]Ga-citrate solution on mAIO^®^ and on iMiDEV™ synthesizers. For the first 30 min on average, the radiotracer was found inside the circulatory system. After 90 min, the tracer was mostly visualized inside the urinary system such as kidneys, ureters, and bladders for its natural elimination. As our animals were all healthy with no inflammation site, [^68^Ga]Ga-citrate, synthetized on mAIO^®^ and on iMiDEV™, was normally distributed in vivo without any suspicious accumulation inside any organ other than the uptake usually observed with this molecule (heart, liver, lungs, urinary tract). (Figure 6)

## 3. Discussion

In this research, we developed the [^68^Ga]Ga-citrate production using a strategy adapted from Jensen et al. on commercially available synthesizers using two different technologies [20]. The first one, mAIO^®^, is well known to realize ^68^Ga-radiosynthesis using a customizable disposable kit allowing macrofluidic production to be performed, whereas the second one, iMiDEV™, is a new commercialized module designed for microfluidic radiosynthesis applying a disposable single-use cassette.

The optimization of the [^68^Ga]Ga-citrate radiosynthesis showed that the nature of cation exchange solid support has a crucial impact on the ^68^Ga^3+^ trap–release property. Six commercial cationic exchange beads were tested for their capacity to trap the ^68^Ga/^68^Ge generator eluate and to desorb ^68^Ga with sodium citrate solution. The trapping was shown to be more efficient with a cation exchange support loaded with sulfonate groups (MCX, PS-H+, AG 50W-X8, and SCX) compared with supports grafted with carboxylate groups (WCX, Accell Plus CM). Our results for Accell Plus CM cartridge contradict the previous study published by Rizello and co-workers, reporting an average 64% yield of [^68^Ga]Ga-citrate, suggesting an efficient level of Ga-68 trapping on the resin [17].

Among the four sulfonate cationic exchangers, MCX and PS-H+ supports demonstrated higher recovery of [^68^Ga]Ga-citrate (up to 85%) compared with AG 50W-X8 and SCX cartridges possessing less than 25% of the recovery efficiency (Table 1). This finding differs from Jensen’s method using SCX to perform [^68^Ga]Ga-citrate production. Nevertheless, SCX is often used to perform purification of ^68^Ga eluate obtained from the ^68^Ge/^68^Ga generator by the Muller method (5 M NaCl/HCl) [27,28]. AG 50W-X8 support was also reported for the purification of ^68^Ga eluate using the Zhernosekov method (HCl/acetone) for ^68^Ga elution [29]. The difference in the desorption efficiency between the chloride used in the Muller and Zhernosekov methods and the citrate chelator can be explained by weak chelating properties of the latter.

As it was shown, the highest radiochemical yields and radiochemical purities were obtained by using a solution of 136 mM sodium citrate (95.5 ± 0.4% for MCX and 88.2 ± 0.3% for PS-H+, Table 2). In contrast, when a mixture of sodium citrate/citric acid (ACD-A) was used for Ga-68 desorption from PS-H+ cartridge, a drastic drop in radiochemical yield was observed (53.0 ± 11.9%, Table 2). This could be explained by the difference in pH of the different citrate solutions (pH = 5 for ACD-A, pH = 5.8 for 74.8 mM sodium citrate, pH = 6.2 for 136 mM sodium citrate). The more basic pH of 74.8 mM and 136 mM sodium citrate solutions seems to be more favourable to desorb Ga-68 from beads in [^68^Ga]Ga-citrate form because, in this pH range, nearly all carboxylic groups of citrate are in carboxylate forms (pKa of 3.13, 4.76, and 6.40 respectively) [30]. These optimizations of the conventional radiosynthesis of [^68^Ga]Ga-citrate using mAIO^®^ resulted in the validation of the production by realizing three consecutive batches.

In contrast with mAIO^®^, the comparison between the trapping–recovery properties of MCX and PS-H+ beads integrated in R1/R3 chambers of the iMiDEV^TM^ microfluidic cassette revealed slightly better results for PS-H+ resin. A small decrease in radiochemical purity of the final [^68^Ga]Ga-citrate solution while using MCX was observed. Regarding the difference in microfluidic network used in the case of passing a fluid either through the R1 or R3 chamber, the comparison of radiochemical yields was performed. The elution with a 136 mM sodium citrate solution through R1 chamber was shown to provide the higher radiochemical yields compared with R3 under the same conditions. Overall, this led us to choose the cassettes with PS-H+ beads integrated in the R1 chamber as the standard for validation of [^68^Ga]Ga-citrate production.

The residual activity distribution was visualized using CLI. An activity loss around 10% was detected to be mainly localized in the beginning of R1 chamber of the cassette. This could be explained by the fact that the Ga-68 capture and elution were performed in the forward direction, contrary to the conventional method in which only about 5% of the loss was determined, probably due to the reverse direction of the elution.

The data obtained from AAS analyses showed the decreased level of Zn (II) in [^68^Ga]Ga-citrate samples compared with the concentration determined in the generator eluate. Moreover, PS-H+ solid support seems to be more efficient for zinc removal compared with MCX. A high Zn concentration could be explained by the zinc formation during Ga-68 decay, regarding the use of the older generator for these measurements. The slightly increased concentration of iron in both mAIO^®^ and iMiDEV^TM^ samples compared with the generator level is probably related to the contamination coming from sealing parts and reagent impurities. These results demonstrate the interest of our strategy in using solid support for the reduction in Zn impurity.

In vitro evaluation using an instant radio-TLC showed that [^68^Ga]Ga-citrate was stable on the shelf for up to 3 h and in serum for up to 1 h (RCP = 82%).

Furthermore, the relatively high SPB and BCB values were observed demonstrating the lipophilic characteristics of the tracer. This aspect potentially facilitates tissue penetration of the radiopharmaceutical, thus allowing a better visualization of infection/inflammation by PET imaging. These high levels are in accordance with the iron-mimetic character of Ga-68 which partially replaces Fe^3+^ cations from serum transport proteins (transferrin, ferritin, and other siderophores) and red blood cell haemoglobin.

Our in vivo study showed a normal distribution of [^68^Ga]Ga-citrate synthesized with mAIO^®^ and iMiDEV^TM^ modules within the organs of the healthy animals. The mean SUV values obtained for [^68^Ga]Ga-citrate synthesized on the conventional module and those produced using the microfluidic system were similar, as both ways of production allow the tracer with the same radiochemical properties (volumic activity, radiochemical purity) to be obtained. An accumulation was observed in the heart, liver, and lungs, as was already demonstrated in other works [19,31,32,33]. Its retention in the blood pool and liver can also be explained, as it is known that ^68^Ga can bind iron transport proteins [21,31].

Thus, the described automatic conventional and microfluidic module-based synthetic methods are able to produce a cGMP [^68^Ga]Ga-citrate in high RCY and RCP, with a suitable on-the-shelf and in vitro stability and with a normal in vivo biodistribution profile.

## 4. Materials and Methods

### 4.1. Radiosynthesis Process

[^68^Ga]Ga-citrate was synthesized using an original and fully automated in-house reaction sequence described in the workflow, Figure 7.

This workflow was implemented for the production of [^68^Ga]Ga-citrate on mAIO^®^ and iMiDEV™ modules. The synthesis cassettes for mAIO^®^ were in-house assembled using manifolds and tubing acquired from TRASIS (Ans, Belgium). The microfluidic cassettes for iMiDEV™ were fabricated by miniFAB (Melbourne, Australia).

The ^68^Ge/^68^Ga GMP-generator was obtained from Eckert and Ziegler (Galliapharm, Berlin, Germany). Cartridges and beads were purchased from Waters, Macherey Nagel, SPure or Bio-Rad) and were used without any preactivation, except for AG 50W-X8, which was previously rinsed with 5 mL of water. Solvents and other reagents were acquired from Merck (Darmstadt, Germany) or Sigma-Aldrich Sweden (Stockholm, Sweden) in high chemical degrees and metal-free media.

### 4.2. Radiosynthesis of [^68^Ga]Ga-Citrate on mAIO^®^ Module

The mAIO^®^ synthesis module is a versatile synthesizer equipped with two arrays of six rotary actuators having multiple positions, one heater, and three radiation detectors, allowing radiosynthesis to be monitored. mAIO^®^ operates with home-made single-use cassettes (structured with three-way valve manifolds) and reagent sets. This facilitates GMP compliance and prevents cross-contamination. The parts available in bulk allow the assembly of these cassettes on which various components can be plugged: spikes (for reagent vials), SPE cartridges, syringes, or tubes. The user-friendly software reproduces the picture of the cassette layout and shows in real time all the movements of the machine during the radiotracer production.

The main steps of [^68^Ga]Ga-citrate radiosynthesis are described as follows (Figure 2): 5 mL of HCl 0.1M (B) is passed through ^68^Ge/^68^Ga generator from SA2 to SA1 at 2 mL/min eluting ^68^Ga in ^68^GaCl_3_ form (5 min). ^68^Ga^3+^ cation (in SA1) is then transferred onto a cation exchange cartridge (P3, P11, P12, P10, Vacuum at −500 mbar, 5 mL/min) in which ^68^Ga^3+^ is trapped. The untrapped ^68^Ga^3+^ is collected in the waste vial. The cartridge is washed with 3 mL of NaCl 0.9% (C–P6) (SA1–P11, P12, P10, Vacuum at −150 mbar, 20 mL/min) and flushed by a nitrogen gas flow (500 mbar; −150 mbar for 30 s). An amount of 4 mL of citrate solution (A, P4) is recovered in SA1 (P3) and passed through the cation exchange cartridge at 20 mL/min rate and flushed by the nitrogen gas flow (500 mBar; −150 mbar for 30 s). [^68^Ga]Ga-citrate is recovered into the product vial after final sterilizing filtration.

The trapping efficiency is calculated as the ratio between the trapped activity on the cartridge and the calculated initial activity of ^68^Ga injected in the module. The recovery efficiency is calculated as the ratio between the eluted activity and the remaining activity trapped on the cartridge after washing. The radiochemical yield is calculated as the ratio between the recovered activity after the elution step and the activity eluted from ^68^Ge/^68^Ga generator.

### 4.3. Radiosynthesis of [^68^Ga]Ga-Citrate on iMiDEV™

iMiDEV™ from PMB-ALCEN (Peynier, France) is a research and development synthesizer for microfluidic radiopharmaceutical production. The iMiDEV™ system manipulates microfluidic cassette with a unique architecture which allows the radiopharmaceutical production panel to be developed. The iMiDEV™ system consists of several subunits to carry out all steps of the synthesis and is controlled by dedicated software.

Disposable microfluidic cassettes for [^68^Ga]Ga-citrate production contain MCX and PS-H+ beads integrated into R1 and R3 chambers. MCX beads were obtained from cartridges purchased from Waters™ and PS-H+ from Macherey-Nagel™.

The radiation detectors close to R1 and R3 were used for trap–release monitoring.

For a single production using R1, vials containing 1 mL of saline solution and 4 mL of 136 mmol sodium citrate solution were placed on positions A and B, respectively (Figure 3). The production started from ^68^GaCl_3_ trapping. The starting activity ranged between 80 and 500 MBq. ^68^Ga-solution was transferred trough injection port and trapped onto the cationic exchange beads (PS-H+ or MCX) integrated in R1 (isotope vial, 1–5 mL, by opening microfluidic valves 6 and 3 towards the waste port using nitrogen flow at 1 bar for 3 min), and the untrapped Ga-68 was collected in the waste vial. Then, washing solution was passed through R1 to the waste vial (Vial A, 1 mL, by opening microfluidic valves 6 and 2 towards the waste port by passing nitrogen at 1 bar for 2 min). Lastly, [^68^Ga]Ga-citrate was formed during elution-formulation step with 136 mmol sodium citrate solution from the R1 chamber (Vial B, 4 mL, by opening microfluidic valves 33, 29, 22, 18, 13, 12, 7, and 1 towards the formulation chamber using nitrogen gas at 1.5 bar for 3 min). The final product was collected from the formulation chamber and was passed through the syringe port and a 0.22 µm sterile filter into the collection vial. Following a similar workflow and using identical time-pressure parameters, [^68^Ga]Ga-citrate was successfully synthesized on the R3 chamber. The cassette loading for the synthesis on R3 included a vial with washing solution (saline, 1 mL) placed at position C and the eluent vial (136 mmol sodium citrate, 5 mL) placed at position G (Figure 3). A detailed description of both procedures is depicted in Appendix A.

The trapping efficiency was calculated as the ratio between the trapped activity on the cartridge and the initial activity of ^68^Ga injected in the module. The recovery efficiency was calculated as the ratio between the eluted activity and the remaining activity trapped on the cartridge after washing. The radiochemical yield was calculated as the ratio between the recovered activity after the elution step and the activity injected in the module at the start of synthesis (SOS) and was decay corrected to the SOS.

### 4.4. Cerenkov Luminescence Imaging (CLI)

The presence and the distribution of the residual activity were verified using Cerenkov luminescence imaging of the cassette at the end of synthesis. Images were acquired from the top and bottom transparent sides of the cassette on an OptiMAX multimodal imaging device (Precision X-Ray inc., North Brandford, CT). The latter consists of a super-cooled optical camera placed into the chamber of a X-RAD 320 preclinical irradiator (Precision X-ray Inc., North Branford, CT, USA). Cerenkov image was acquired with a 4 min acquisition duration and an aperture of 0.5. To accurately localize potential foci, a white-light image was also performed to observe the cassette (acquisition duration = 0.5 s, aperture = 0.9). A merging of Cerenkov and white-light images (Figure 4) was applied using ImageJ software. The white-light image was optimized to provide a better visualization.

### 4.5. Quality Control Procedure

Parameters, methods, and specifications required for the radiosynthesis batch validation are summarized in Table 7.

The appearance was determined by visual inspection of batch behind a lead-shielded glass window. The solution must be clear, colourless, and free of particles.

The pH of the solution was determined by applying a drop of solution onto a pH indicator strip and comparing the colour with the provided scale. The value was expected to range between 4.5 and 8.5.

The radiochemical purity was determined by thin layer chromatography (TLC) on iTLC-SG strips as stationary phase eluted with (1) sodium acetate (1.36 g) + acetic acid (0.58 mL) in 100 mL water; (2) ACD-A, using Mini GITA from Elysia-Raytest^®^ as TLC scanner. The radiochemical purity was evaluated by a radioactive chromatogram analysis, and the limit was fixed to be superior to 95% [34].

The radiochemical identity was determined by analytical high performance liquid chromatography (HPLC) analyses with UV and radioactivity detections by comparing the retention times of standard Ga-citrate purchased from ABX (Radeberg, Germany) and [^68^Ga]Ga-citrate. Final [^68^Ga]Ga-citrate product (20 µL) and reconstituted Ga-citrate at 1 mg/mL in water were injected on HPLC run on a Waters Alliance e2695 system equipped with a 2998 photodiode array (PDA) detector and the radio HPLC detector (Herm LB500 with NaI detector, Berthold, Bad Wildbad, Germany) controlled by the Empower Software (Orlando, FL, USA). A Vydac 218TP C-18 250 × 4.6 mm (5 µm, GRACE) analytical column was used, and a multi-step gradient of 1 mL/min was applied using solvent A (potassium phosphate 13.6 g/L), solvent B (0.1% TFA in water), and solvent C (0.1% TFA in acetonitrile): 100% solvent A; 5–10 min: from 100% solvent A to 95% solvent B/5% solvent C; 10–15 min: 95% solvent B/5% solvent C; 15–20 min: from 95% solvent B/5% solvent C to 100% solvent A.

The radionuclide purity and identity of [^68^Ga]Ga-citrate were determined by gamma-ray spectrometry (miniGITA, Elysia-Raytest^®^, Straubenhardt, Germany) and a time-decay method using the dose calibrator (CRC^®^-25R, Capintec, Inc., Florham Park, NJ, USA). The radionuclide purity [^68^Ga]Ga-citrate was evaluated by an identification of 511 keV gamma-ray peak (>99%). The radionuclide identity was confirmed measuring a half-life ranging between 62 and 74 min [34].

The sterility and the bacterial endotoxin contamination were tested on the [^68^Ga]Ga-citrate samples (post radioactive decay) of each batch according to the *European Pharmacopoeia* monograph requirements. The sterility was verified by a direct inoculation of at least 1 mL of [^68^Ga]Ga-citrate with resazurin thioglycolate and trypticase soya media. The culture tubes were incubated at 32.5 °C for resazurin thioglycolate medium and at 22.5 °C for trypticase soya medium for 14 days and visually daily inspected. The analysis of the endotoxin content in the [^68^Ga]Ga-citrate solution was performed using rapid test cartridges for LAL (Endosafe^®^ NexGen-PTS™, Charles River).

### 4.6. Quality Control of Metal Ions Impurities

The trace level detection of the metal ion contamination in decayed samples was carried out by atomic absorption spectrometry (AAS) using flame (F AAS) and graphite furnace (GF AAS) systems (LIEC, University of Lorraine, Nancy, France). The water used was Milli-Q purity (Millipore, Burlington, MA, USA), and the acids were purified by sub-boiling distillation using a duoPUR system (Milestone, Brondby, Denmark). The non-disposable materials used were previously washed with 5% HNO_3_ and distilled water. Quantification was carried out by external calibration and internal standard, preparing the standards by successive dilution of a certified standard (Alfa Aesar (Haverhill, MA, USA) and Inorganic Ventures) to 2–5% (v/v) using 65% HNO_3_ (Sigma-Aldrich, St. Louis, MO, USA for each of the elements.

### 4.7. In Vitro Stability

The stability of [^68^Ga]Ga-citrate was evaluated in final formulated media and in serum. Stability in final formulated solution kept at room temperature was determined by radio-TLC analyses at the end of synthesis, 1 h, 2 h, and 3 h post-radiolabelling.

The stability in serum was evaluated as follows: an aliquot of 50 µL [^68^Ga]Ga-citrate was added to 450 μL of serum, and the mixture was incubated at 37 °C under slight agitation. After 1 h post-incubation, solution was treated with 500 µL of acetonitrile to precipitate the proteins, vortexed for 1 min, and then centrifuged (2500 rpm, 5 min) with ALC^®^ PK131R (Analis—Namur, Belgium). The supernatant samples were analysed with radio-TLC.

### 4.8. Serum Protein Binding (SPB) Evaluation

The SPB of [^68^Ga]Ga-citrate was evaluated as described elsewhere [35]. Briefly, an aliquot of [^68^Ga]Ga-citrate (50 μL/~0.5 MBq) was incubated at 37 °C for 1 h with 450 μL of rat serum (*n* = 3) under slight agitation. After the incubation period, serum proteins were precipitated with acetonitrile (1:1), vortexed during 1 min, and the samples were centrifuged (2500 rpm, 5 min) with ALC^®^ PK131R (Analis—Namur, Belgium). The activities of the pellets and supernatants were measured in an automatic gamma counter (2470 Wizard2^TM^, PerkinElmer, Switzerland). The SPB was calculated as follows:(1)$$\mathrm{SPB}=\frac{\mathrm{cpm}\;\left(\mathrm{pellet}\right)}{\mathrm{cpm}\;\left(\mathrm{pellet}+\mathrm{supernatant}\right)}\times 100$$

### 4.9. Blood Cell Binding

Briefly, an aliquot of [^68^Ga]Ga-citrate (50 μL/~0.5 MBq) was incubated at 37 °C for 1 h with 450 μL of rat blood (*n* = 3) under slight agitation. After the incubation period, samples were centrifuged (2500 rpm, 5 min) with ALC^®^ PK131R (Analis—Namur, Belgium) to determine the activity binding with blood cells. The activities of the pellets and supernatants were measured in an automatic gamma counter (2470 Wizard2^TM^, PerkinElmer, Switzerland). The BCB was calculated as follows:(2)$$\mathrm{BCB}=\frac{\mathrm{cpm}\;\left(\mathrm{pellet}\right)}{\mathrm{cpm}\;\left(\mathrm{pellet}+\mathrm{supernatant}\right)}\times 100$$

### 4.10. In Vivo Analysis of [^68^Ga]Ga-Citrate Biodistribution Using MicroPET

The organism distribution of [^68^Ga]Ga-citrate was evaluated in 3 healthy Wistar-Han breed rats, weighting between 250 and 300 g. The PET acquisition was performed on a small animal-dedicated PET system (Inveon, Siemens, Knoxville, TN, USA). The animals were anesthetized by inhalation of isoflurane and received 26.7 ± 0.5 MBq of [^68^Ga]Ga-citrate in a volume of 0.2 ± 0.02 mL by I.V. injection. The animals were placed in prone position on a heating pad. The PET acquisition was started at the time of the radiotracer injection with a recording time of 120 min for ^68^Ga-emission and 30 min for ^57^Co transmission. Images were reconstructed in a dynamic of 5 frames of 2 min and 22 frames of 5 min using the ordered subsets expectation maximization 3D algorithm (OSEM3D) together with scatter and attenuation corrections based on transmission source measurement. The final voxel size was 0.8 × 0.8 × 0.9 mm^3^. A region of interest (ROI) was placed on several organs, such as liver, heart, kidneys, bladder, bone marrow, and bone. The corresponding volume of interest was applied to the full dynamic data set to generate a time–activity curve (TAC). This animal experiment was conducted in accordance with protocols approved by the Lorraine Ethic Committee n°66 according to guidelines of animal care and use.

Standard uptake value (SUV) means of the various organs obtained with [^68^Ga]Ga-citrate were compared for the product synthetized on the two modules mAIO^®^ and iMiDEV™ using multiple unpaired *t*-tests (GraphPad Prism 9.2.0).

## 5. Conclusions

This study demonstrates the possibility of a robust and reliable production of [^68^Ga]Ga-citrate on conventional mAIO^®^ module as well as on microfluidic iMiDEV™ disposable cassette via an efficient and fast trap–release from solid support in high reproducible yields and purities. In order to respond to the often-overestimated requirements of regulatory authorities, a well-implemented GMP compliant quality control procedure was defined. For the measurements of radiochemical purity, the most appropriate radio-TLC method was implemented after screening of multiple protocols found in the literature. Moreover, a new method of radiochemical identity determination was validated by using reverse phase HPLC analysis. Similar biodistribution estimated using micro-PET imaging on healthy rats confirmed uniformity between macro- and microfluidic technologies applied for [^68^Ga]Ga-citrate production purposes. Our results could be prominent to extending both the production (mAIO^®^ or iMiDEV™) and clinical application of this radiopharmaceutical.

## Figures and Tables

**Figure 1 molecules-27-00994-f001:**
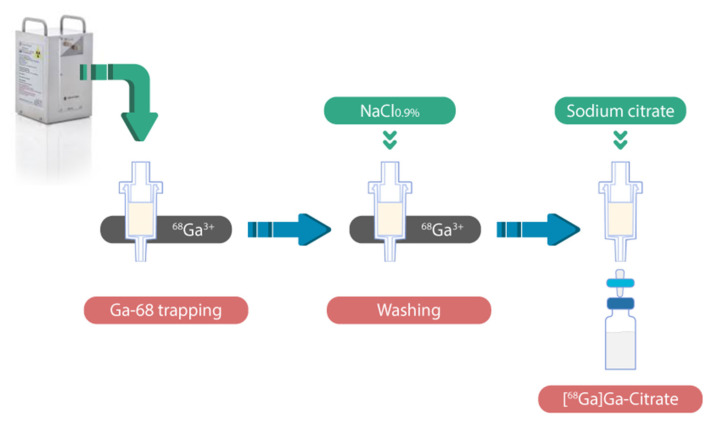
[^68^Ga]Ga-citrate radiosynthesis strategy.

**Figure 2 molecules-27-00994-f002:**
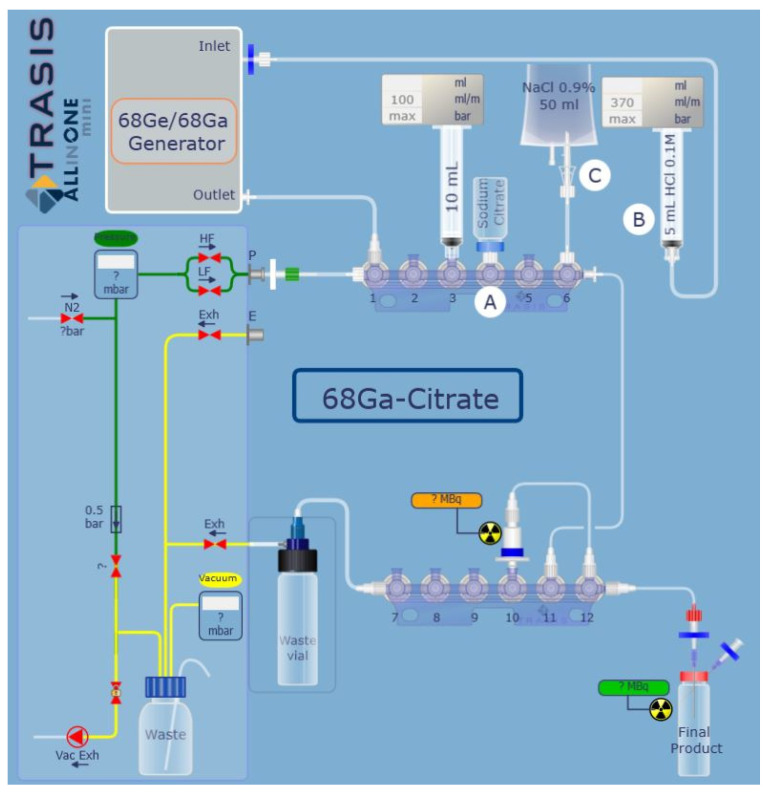
Graphical representation of radiosynthesis of [^68^Ga]Ga-citrate on mAIO^®^ module.

**Figure 3 molecules-27-00994-f003:**
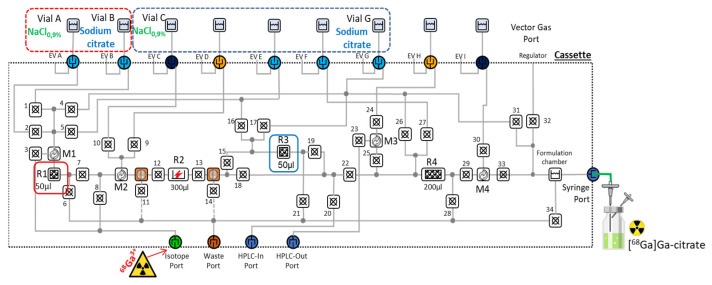
Schematic representation of iMiDEV™ microfluidic cassette used for [^68^Ga]Ga-citrate production; highlighted in red—single synthesis on R1 with Vials A and B; highlighted in blue—single synthesis on R3 with vials C and G.

**Figure 4 molecules-27-00994-f004:**
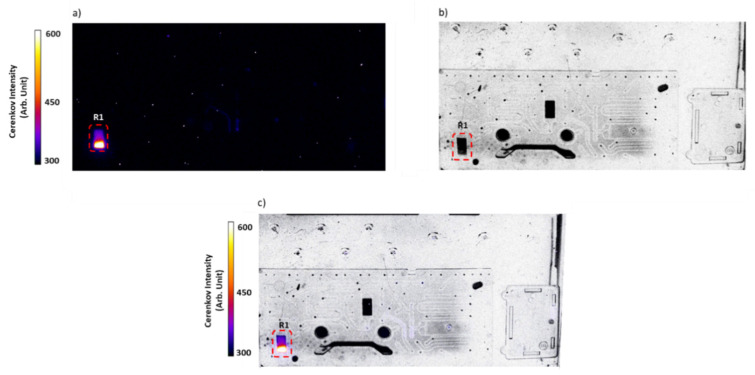
Cerenkov imaging of the residual activity distribution on the cassette after single production of [^68^Ga]Ga-citrate on R1 chamber with PS-H+ beads; (**a**) Cerenkov image after synthesis; (**b**) white-light image of the top of the cassette; (**c**) merged Cerenkov and white-light images after completion of the synthesis.

**Figure 5 molecules-27-00994-f005:**
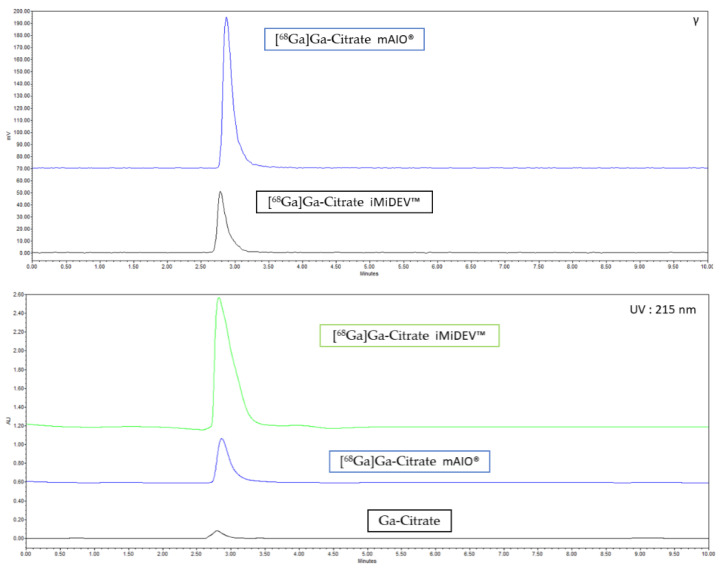
HPLC chromatograms of [^68^Ga]Ga-citrate production. (**Top**) radioactive detection; (**Bottom**) UV 215 nm detection.

**Figure 6 molecules-27-00994-f006:**
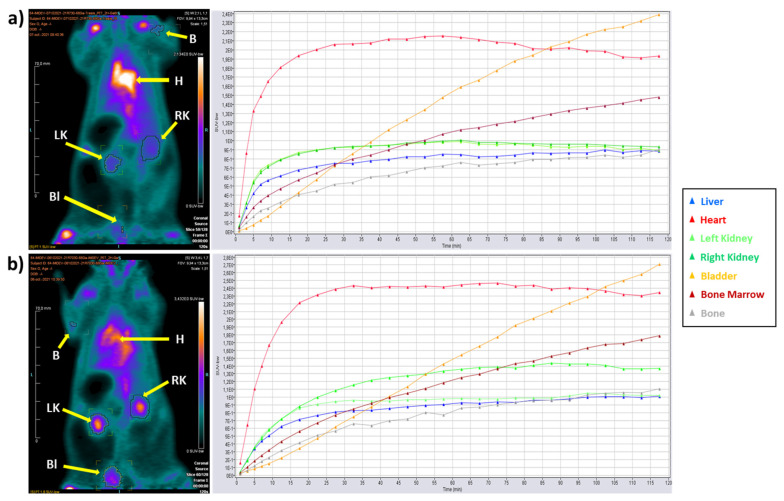
Dynamic µPET imaging of [^68^Ga]Ga-citrate up to 120 min post injection (coronal slice; injected dose: 26.7 ± 0.5 MBq; scan duration: 5 frames of 2 min and 22 frames of 5 min); (**a**) PET image and time activity curve (TAC) of [^68^Ga]Ga-citrate synthesized on mAIO^®^; (**b**) PET image and TAC of [^68^Ga]Ga-citrate synthesized on iMiDEV™. B: bone; H: heart; LK: left kidney; RK: right kidney; Bl: bladder.

**Figure 7 molecules-27-00994-f007:**
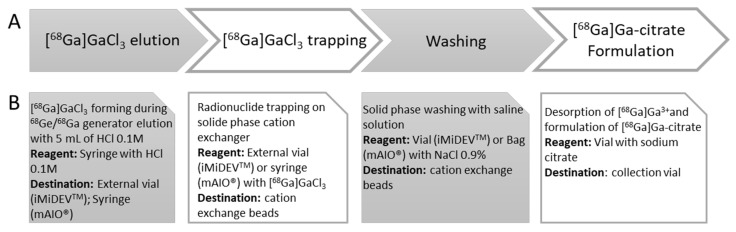
Workflow for automated [^68^Ga]Ga-citrate production using miniAIO^®^ and iMiDEV™ modules. (**A**) Overview of main steps of [^68^Ga]Ga-citrate production. (**B**) Card panels represent basic processes needed to achieve each of main steps of production.

**Table 1 molecules-27-00994-t001:** Comparison of ^68^Ga^3+^ trapping and recovery efficiencies with 136 mM solution of sodium citrate from different solid supports (*n* = 3).

Type of Cartridge	Trapping Efficiency (%)	Recovery Efficiency (%)
WCX	3.66 ± 0.5	ND
Accell plus CM	0.32 ± 0.2	ND
MCX	99.50 ± 0.1	96.13 ± 0.2
PS-H+	99.23 ± 0.1	88.92 ± 0.8
AG 50W-X8 *	99.70 ± 0.1	22.75 ± 0.5
SCX Maxi-Clean	99.60 ± 0.4	24.04 ± 1.1

ND—not determined. * AG 50W-X8 cartridge—in-house prepared with 600 mg of commercial resin.

**Table 2 molecules-27-00994-t002:** Influence of the eluent solution on the radiochemical yield using two types of solid support (PS-H+ and MCX). Volume of eluant was fixed at 4 mL (*n* = 3).

Type of Cartridge	Eluent	RCY (%)	RCP (%)
PS-H+	ACD formula A	53.0 ± 11.9	100
	Sodium Citrate 74.8 mM	71.1 ± 2.7	100
	Sodium Citrate 136 mM	88.2 ± 0.7	99.9 ± 0.1
MCX	ACD formula A	90.4 ± 1.0	100
	Sodium Citrate 74.8 mM	94.7 ± 0.5	100
	Sodium Citrate 136 mM	95.5 ± 0.4	100

RCY—Radiochemical yields are decay corrected to the start of elution (SOE); RCP—radiochemical purity measured by radio-TLC.

**Table 3 molecules-27-00994-t003:** Comparison of trapping and recovery efficiency of ^68^Ga^3+^ using different chambers and SPE resins.

Type of Cartridge	Trapping R1 (%)	Trapping R3 (%)	Recovery R1 (%)	Recovery R3 (%)
MCX	95.3 ± 2.7 (*n* = 3)	96.2 ± 2.8 (*n* = 3)	89.3 ± 0.8 (*n* = 2)	86.4 ± 0.2 (*n* = 2)
PS-H+	98.8 ± 1.1 (*n* = 9)	98.9 ± 0.4 (*n* = 5)	90.5 ± 2.6 (*n* = 7)	88.5 ± 6.3 (*n* = 4)

**Table 4 molecules-27-00994-t004:** Radiochemical yields, purity, and residual activity values for [^68^Ga]Ga-citrate synthesis.

Chamber	PS-H+			MCX		
	RCY (dc), %	Residual Activity on the Cassette (dc), %	RCP, %	RCY (dc), %	Residual Activity on the Cassette (dc), %	RCP, %
R1	90.1 ± 3.1 (*n* = 6)	9.2 ± 2.7 (*n* = 6)	98.7 ± 1.2 (*n* = 6)	84.9 ± 4.8 (*n* = 2)	10.2 ± 0.3 (*n* = 2)	98.8 ± 3.2 (*n* = 2)
R3	87.5 ± 5.5 (*n* = 5)	10.3 ± 5.8 (*n* = 5)	99.3 ± 1.0 (*n* = 5)	80.7 ± 0.5 (*n* = 2)	12.7 ± 0.3 (*n* = 2)	89.5 ± 8.5 (*n* = 2)

RCY—Radiochemical yields are decay corrected to the start of synthesis (SOS). RCP—Radiochemical purity estimated from radio-TLC analyses.

**Table 5 molecules-27-00994-t005:** Summary of concentrations of the metallic impurities of the ^68^Ge/^68^Ga generator and in the [^68^Ga]Ga-citrate samples atomic absorption data (*n* = 2).

Metal	Blank HCl 0.1 M/136 mM Sodium Citrate µg/L	^68^Ge/^68^Ga Generator GalliaPharm µg/L	[^68^Ga]Ga-Citrate mAIO^®^ µg/L	[^68^Ga]Ga-Citrate iMiDEV™ µg/L
Zn	7.45/12.43	39,010.8 ± 2378.4	8046.9 ± 3687.3	235.9 ± 38.8
Fe	2.23/33.03	22.4 ± 0.1	85.4 ± 53.6	128.3 ± 18.0

**Table 6 molecules-27-00994-t006:** Comparison of chromatographic properties of different systems tested for [^68^Ga]Ga-citrate radiochemical purity measurements.

References	Stationary Phase /Mobile Phase	^68^Ga^3+^ (Rf)	[^68^Ga]Ga-Citrate (Rf)
Xu et al. [6]	Whatman/ Methanol/Ammonium acetate 10% (1:1)	0.1	0.2
Jensen et al. [20]	iTLC-SG/ Sodium acetate (1.36 g) + acetic acid (0.58 mL) in 100 mL Water	0.1	0.9
Mirzaie et al. [21]	Whatman/ Sodium acetate (1.5 g) + acetic acid (0.58 mL) in 100 mL Water	0.2/0.95	0.95
Aghanajab et al. [5]	Whatman/ Acetone/glacial acetic acid (3:1)	0.3	1
Rizetto et al. [17]	iTLC-SG/ Methanol/ acetic acid (9:1)	0	1
-	iTLC-SG/ ACD	1	1

**Table 7 molecules-27-00994-t007:** Results of validation of [^68^Ga]Ga-citrate production with three consecutive batches.

Parameter	Method (Criteria of Acceptance)	mAIO^®^ Production	iMiDEV™ Production
Radiosynthesis time	-	11 min from SOE	10 min from SOS
Radiosynthesis Yield	-	96.1 ± 0.5 %	90.1 ± 2.6%
Appearance	Visual inspection (colourless)	passed	passed
pH	pH strips (4–8)	6	7
Radiochemical purity	Radio-TLC (>95%)	99.8 ± 0.1%	98.5 ± 1.3%
Radionuclide purity	Gamma-spectrometry (511 keV +/− 10%)	532 ± 3 keV	532 ± 3 keV
Radionuclide identity	Half-life (62–74 min)	69.3 min	69.9 min
Bacterial endotoxins	LAL test (UE < 75/mL)	<1.5 UE/mL	<1.5 UE/mL
Sterility		sterile	sterile

**Table 8 molecules-27-00994-t008:** Mean standard uptake values (SUV_mean_) by body weight (bw) of [^68^Ga]Ga-citrate in various organs in healthy rats.

Organs	SUV_mean_/bw [^68^Ga]Ga-Citrate mAIO^®^	SUV_mean_/bw [^68^Ga]Ga-Citrate iMiDEV™
Liver	0.95 ± 0.31	0.83 ± 0.18
Heart	2.61 ± 1.24	2.40 ± 0.46
Left Kidney	1.07 ± 0.29	0.97 ± 0.05
Right Kidney	1.23 ± 0.30	1.13 ± 0.06
Bladder	2.30 ± 1.56	1.66 ± 1.14
Bone Marrow	1.20 ± 0.35	1.10 ± 0.29
Bone	0.88 ± 0.23	0.76 ± 0.24

## Data Availability

The data presented in this study are available on request from the corresponding authors.

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
