# Peer review of "Fully Automated Macro- and Microfluidic Production of [68Ga]Ga-Citrate on mAIO® and iMiDEVTM Modules"

_molecules, 2022, doi:10.3390/molecules27030994_

Round 1

Reviewer 1 Report

Very interesting publication, the teams have a wide skill  in radioactivity, radiochemistry and preclinical studies.
The methodology is mastered, each element is explained starting from a bibliographic reference.

I noticed a small error (line 80), I think it should read macrofluidic and not microfluidic.
Concerning the subject, I would have liked to see the behavior of gallium-68 citrate  at high activity (>500 MBq) and especially > 1 000 MBq especially in terms of quality control
This could have an influence on the stability of gallium-68 citrate and also on the colloid rate. 
In the micropet imaging part, I would have liked to see an 18FDG image in comparison, to show the interest of this tracer
good  job

Author Response

Thank you for your review.

I noticed a small error (line 80), I think it should read macrofluidic and not microfluidic.

Response:

Thank you. It was corrected. 

Concerning the subject, I would have liked to see the behavior of gallium-68 citrate  at high activity (>500 MBq) and especially > 1 000 MBq especially in terms of quality control
This could have an influence on the stability of gallium-68 citrate and also on the colloid rate. 

Response:

We agree with the reviewer. However, to have access to activity > 1000 MBq, the 68Ge/68Ga generator need to be used in the first months after its purchase. Unfortunately, this series of experiments were performed when 68Ge/68Ga generator already had decayed to activity < 1000 MBq.

In the micropet imaging part, I would have liked to see an 18FDG image in comparison, to show the interest of this tracer.

Response:

In the present work, we didn’t show any comparison 18F-FDG and 68Ga-Citrate as we only had presented results on healthy animals. In a further work, we have the ambition to demonstrate the interest of this radiotracer on animals with an inflammatory condition by comparing this tracer with 18F-FDG.

Reviewer 2 Report

The authors are to be congratulated on the presentation of some valuable and unexpected observations on the clinical production of 68Ga Citrate. The ability to use other production platforms to this agent will prove invaluable. I believe that this will also stir more interest in using microfluidics for clinical production of other agents. 

In its current form the manuscript will require considerable editing to make it more readable. It is mostly English usage rather than scientific content. I have provided extensive suggestions below from the first page alone which I believe are representative of the whole document. Unfortunately time constraints prevent me from making similar listings for all pages. It really requires someone familiar with the work to provide a better English rendition.  It is not my intention to completely remove the "French Flavor" of the language.

Before proceeding to suggestions for change I would like the authors to include a reason for describing the use of both R1 and R3 reactors in the microfluidic syntheses. It is not clear to me the rationale for this.

I will go through the first page line by line in order with my suggestions:

Line  11. Replace "showed" with "has been shown"

Line  12. After "cardiovascular" add "systems"

Line  16. before "radiopharmaceutical " add "a"

Line  17. Change "in complement to" to "to complement"

Line  18. Change "MCX cartridge on conventional" to "a MCX cartridge on the conventional"

Line  19. Insert ", mAIO," between "module " and "while"; insert "a" between "while and PS-H"; and insert "the" between "into and microfluidic

Line  21. Delete "was" and "in" to give "synthesizers demonstrated reliable"

Line  29. Between "to radioisotope" insert "the gallium"

Line  31. Delete "the" in "to the imaging"

Line  34. Replace "his" with "its"

 Line  36. Add "s"  to "agent"' and "receptor"

Line  39. Add "in" between "diagnosis and clinical" AND "s" after "application"

Line  40. Change to "It has been used for imaging inflammation and infection by SPECT for ...."

Line  41. I note here that references #1through #3 do not refer to Ga-67 specifically but rather to Ga-68! 

Line  43.  Delete "interest"

line  44. Change "diagnosis" to "diagnose"

Line  44. insert "(" before "two"

Author Response

Thank you for your review.

In its current form the manuscript will require considerable editing to make it more readable. It is mostly English usage rather than scientific content. I have provided extensive suggestions below from the first page alone which I believe are representative of the whole document. Unfortunately time constraints prevent me from making similar listings for all pages. It really requires someone familiar with the work to provide a better English rendition.  It is not my intention to completely remove the "French Flavor" of the language.

Response:

The manuscript was revised to improve English.

Before proceeding to suggestions for change I would like the authors to include a reason for describing the use of both R1 and R3 reactors in the microfluidic syntheses. It is not clear to me the rationale for this.

Response:

Thank you very much for this remark. The information had been added in the manuscript L 171 :

Microchambers R1 and R3 with a same volume of 50 µL were chosen as reaction chambers. These chambers are recommended for filling with SPE beads and could be both used for the radionuclide trapping step. This should allow to perform two productions of [68Ga]Ga-citrate radiopharmaceutical using the same cassette.

I will go through the first page line by line in order with my suggestions:

Line  11. Replace "showed" with "has been shown"

Line  12. After "cardiovascular" add "systems"

Line  16. before "radiopharmaceutical " add "a"

Line  17. Change "in complement to" to "to complement"

Line  18. Change "MCX cartridge on conventional" to "a MCX cartridge on the conventional"

Line  19. Insert ", mAIO," between "module " and "while"; insert "a" between "while and PS-H"; and insert "the" between "into and microfluidic

Line  21. Delete "was" and "in" to give "synthesizers demonstrated reliable"

Line  29. Between "to radioisotope" insert "the gallium"

Line  31. Delete "the" in "to the imaging"

Line  34. Replace "his" with "its"

 Line  36. Add "s"  to "agent"' and "receptor"

Line  39. Add "in" between "diagnosis and clinical" AND "s" after "application"

Line  40. Change to "It has been used for imaging inflammation and infection by SPECT for ...."

Line  41. I note here that references #1through #3 do not refer to Ga-67 specifically but rather to Ga-68! 

Line  43.  Delete "interest"

line  44. Change "diagnosis" to "diagnose"

Line  44. insert "(" before "two"

            All corrections were added.

Reviewer 3 Report

The manuscript submitted by Ovdiichuk O, et al. reports automatic production of Ga-68 labeled citrate using a macrofluidic and a microfluidic synthesis module. The parameters relating the synthesis yield (trapping and desorption) were carefully optimized, and the quality of the final Ga-68 citrate formulation for human injection was demonstrated by various in vitro and in vivo studies. This manuscript is well written and the conclusions are supported by the presented data. Therefore, this manuscript could be accepted after the following suggested minor changes have been properly addressed:

  • Line 80: “The first one is the microfluidic miniAIO® (Trasis) have been used for production of numerous 68Ga-radiopharmaceuticals like [68Ga]Ga-PSMA-11 [20,21], [68Ga]Ga-NODAGA-RGD [22].” should be “The first one is the macrofluidic miniAIO® (Trasis) which has been used for production of numerous 68Ga-radiopharmaceuticals like [68Ga]Ga-PSMA-11 [20,21] and [68Ga]Ga-NODAGA-RGD [22].”.
  • Line 190: “A slight decrease of about 7% in radiochemical yield occurred when using R3 chamber filled either with PS-H+ or MCX beads (Table 4).”. This 7% decrease is different from the numbers presented in Table 4 (90.1 vs 87.5 and 84.9 vs 80.7).
  • As presented in Table 5, there are high average concentrations of Zn (8 ppm) and Fe (0.1 ppm). Would these concentrations be too high for injection? What are the acceptable levels for Zn and Fe concentrations in the final formulations?
  • Figure 5: There should be an injection of free Ga-68 for comparison.
  • Figure 6: The organ names for different color lines on the top right corners of Fig 6A and Fig 6B should be enlarged to be legible.

Author Response

Thank you for your review.

  • Line 80: “The first one is the microfluidic miniAIO® (Trasis) have been used for production of numerous 68Ga-radiopharmaceuticals like [68Ga]Ga-PSMA-11 [20,21], [68Ga]Ga-NODAGA-RGD [22].” should be “The first one is the macrofluidic miniAIO® (Trasis) which has been used for production of numerous 68Ga-radiopharmaceuticals like [68Ga]Ga-PSMA-11 [20,21] and [68Ga]Ga-NODAGA-RGD [22].”.

Thank you. It was corrected. 

  • Line 190: “A slight decrease of about 7% in radiochemical yield occurred when using R3 chamber filled either with PS-H+ or MCX beads (Table 4).”. This 7% decrease is different from the numbers presented in Table 4 (90.1 vs 87.5 and 84.9 vs 80.7).

Thank you. It was corrected. It is not 7 % but about 3-4 %

  • As presented in Table 5, there are high average concentrations of Zn (8 ppm) and Fe (0.1 ppm). Would these concentrations be too high for injection? What are the acceptable levels for Zn and Fe concentrations in the final formulations?

Thank you very much for this remark. However, this analysis has been realized on the samples prepared with an “old” decayed 68Ge/68Ga generator which explains the high average concentration of Zn. The purpose of this experiment was to show that our radiosynthesis strategy allows to reduce metals levels thanks to solid support trapping of Ga-68. The concentration of Fe in all samples was found to be lower the acceptance level of 10 µg/GBq indicated in the Monograph - Gallium (68Ga) chloride solution for radiolabelling (as well a zinc level in iMiDEV sample reduced to the concentration < 10 µg/GBq indicated in the same Monograph). It is worth to notice that our validation batches were prepared using a newer generator which limits the risk of Zn breakthrough and no AAS analyses have been performed for these samples.

This information is discussed in the Result section. 2.2

These results demonstrate that our radiosynthesis strategy allows to considerably decrease Zn level in the final formulated solution in comparison with Zn(II) concentration in the generator eluate. The radiosynthesis on the mAIO® module using MCX solid support reveals to be less efficient for zinc removal compare to the radiosynthesis on the iMiDEVTM cassette filled with PSH+ beads. Very low concentrations of Fe were measured in all samples.

And in Discussion section (L377):

The data obtained from AAS analyses has shown the decreased level of Zn (II) in [68Ga]Ga-citrate samples compared to the concentration determined in the generator eluate. Moreover, PS-H+ solid support seems to be more efficient for zinc removal compared to MCX. A high Zn concentration could be explained by the zinc formation during Ga-68 decay regarding the use of older generator for these measurements. The slightly increased concentration of iron in both mAIO® and iMiDEVTM samples compared to the generator level is probably related to the contamination coming from sealing parts and reagents impurities. These results demonstrate the interest of our strategy using solid support for the reduction of Zn impurity.

  • Figure 5: There should be an injection of free Ga-68 for comparison.

Thank you very much for this remark. We didn’t talk about free gallium because we can’t see it under these HPLC conditions. The information was added in the manuscript: L 277

“ Free Ga-68 was injected under the same conditions, but no detection of 68GaCl3 was observed in 10 min analysis

  • Figure 6: The organ names for different color lines on the top right corners of Fig 6A and Fig 6B should be enlarged to be legible.

Thank you. It was modified.